# 3D-Printed Bioactive Scaffold Loaded with GW9508 Promotes Critical-Size Bone Defect Repair by Regulating Intracellular Metabolism

**DOI:** 10.3390/bioengineering10050535

**Published:** 2023-04-27

**Authors:** Fangli Huang, Xiao Liu, Xihong Fu, Yan Chen, Dong Jiang, Tingxuan Wang, Rongcheng Hu, Xuenong Zou, Hao Hu, Chun Liu

**Affiliations:** 1Guangdong Provincial Key Laboratory of Orthopaedics and Traumatology, Department of Spinal Surgery, The First Affiliated Hospital of Sun Yat-sen University, Guangzhou 510080, Chinazouxuen@mail.sysu.edu.cn (X.Z.); 2Precision Medicine Institute, The First Affiliated Hospital of Sun Yat-sen University, Guangzhou 510080, China

**Keywords:** 3D-printed scaffold, critical-size bone defect, osteogenesis, metabolomics, GW9508

## Abstract

The process of bone regeneration is complicated, and it is still a major clinical challenge to regenerate critical-size bone defects caused by severe trauma, infection, and tumor resection. Intracellular metabolism has been found to play an important role in the cell fate decision of skeletal progenitor cells. GW9508, a potent agonist of the free fatty acid receptors GPR40 and GPR120, appears to have a dual effect of inhibiting osteoclastogenesis and promoting osteogenesis by regulating intracellular metabolism. Hence, in this study, GW9508 was loaded on a scaffold based on biomimetic construction principles to facilitate the bone regeneration process. Through 3D printing and ion crosslinking, hybrid inorganic-organic implantation scaffolds were obtained after integrating 3D-printed β-TCP/CaSiO_3_ scaffolds with a Col/Alg/HA hydrogel. The 3D-printed β-TCP/CaSiO_3_ scaffolds had an interconnected porous structure that simulated the porous structure and mineral microenvironment of bone, and the hydrogel network shared similar physicochemical properties with the extracellular matrix. The final osteogenic complex was obtained after GW9508 was loaded into the hybrid inorganic-organic scaffold. To investigate the biological effects of the obtained osteogenic complex, in vitro studies and a rat cranial critical-size bone defect model were utilized. Metabolomics analysis was conducted to explore the preliminary mechanism. The results showed that 50 μM GW9508 facilitated osteogenic differentiation by upregulating osteogenic genes, including *Alp*, *Runx2*, *Osterix,* and *Spp1* in vitro. The GW9508-loaded osteogenic complex enhanced osteogenic protein secretion and facilitated new bone formation in vivo. Finally, the results from metabolomics analysis suggested that GW9508 promoted stem cell differentiation and bone formation through multiple intracellular metabolism pathways, including purine and pyrimidine metabolism, amino acid metabolism, glutathione metabolism, and taurine and hypotaurine metabolism. This study provides a new approach to address the challenge of critical-size bone defects.

## 1. Introduction

Critical-size bone defects are a type of severe bone defect in which bone tissue cannot heal spontaneously because of the large scale of the defect [1]. The repair of critical-size bone defects due to common clinical causes, such as high-energy trauma, severe infection, and bone tumors, requires a large amount of bone graft tissue, which is a major clinical challenge in the field of orthopedics [2]. Autologous iliac bone grafting is currently the gold standard for the treatment of critical-size bone defects; however, this has disadvantages, such as donor site injury, limited graft volume, and prolonged operative time. Allogeneic bone and artificial bone (mainly calcium-phosphorus inorganic-based materials) possess good mechanical properties and osteoconductivity, but they usually suffer from a lack of biological activity for inducing new bone formation, and they are only suitable for the repair of small-bone defects. Techniques for bone reconstruction include the induced membrane technique pioneered by Masquelet and the distraction osteogenesis technique proposed by Ilizarov [2,3]. However, the former has the disadvantage of a second surgery, and it still requires a large amount of bone graft tissue, such as autologous bone, while the latter has the disadvantage of being very time-consuming (on average, it requires 10 months) and is poorly accepted by patients. For treating critical-size bone defects, all of the traditional treatments have certain risks and drawbacks, often leading to a series of complex clinical problems, such as poor bone healing, bone nonunion, and failure of internal fixation. Therefore, the clinical challenge of critical-size bone defects remains unresolved, and it is urgent that novel bone grafts are developed with a comprehensive combination of properties, including mechanical and structural properties that match the bone tissue [4], bioadaptability that creates a cell-friendly microenvironment [5], and bioactivity that induces new bone formation [6].

To cope with critical-size bone defects, this study provides a precise and comprehensive design of bone graft material based on biomimetic construction principles [7]. Because bone is a hybrid inorganic-organic mineralized porous matrix, hybrid inorganic-organic scaffolds should be fabricated that include: rigid materials as the mechanical support and scaffolding structures, soft hydrogels that mimic the extracellular matrix (ECM) for cell adhesion and growth [8], and bioactive factors with strong bone-inducing activity.

For rigid materials, 3D-printing techniques, which allow high precision, can be used to build a porous architecture of ceramic scaffolds [9], and they also provide the advantage of fabricating implants for personalized bone defects [10]. Thus, 3D-printed bioactive ceramic β-TCP/CaSiO3 scaffolds were fabricated based on the technological process developed in our previous studies [11,12], with mineral elements (Ca, P, and Si) that favored bone regeneration [13]. For soft materials, a biomimetic hydrogel composed of natural collagen type I, hyaluronic acid, and alginate that shared similar physicochemical properties with the ECM was synthesized based on a technique from our previous study [14].

As for osteogenic bioactive factors, a novel pro-osteogenesis molecule GW9508 was chosen. GW9508 is a potent agonist of the free fatty acid receptors GPR40 and GPR120, and it was originally developed by researchers to regulate insulin secretion [15]. GPR40 and GPR120 are found in many types of cells, including osteoblasts and osteoclasts [16,17]. Thus, in addition to the broad anti-inflammatory [18], anti-senescence [19], and neuroprotection effects [20] of GW9508, it has recently been found to modulate osteoblast and osteoclast function and exert anti-osteoporotic effects. *GPR40* knockout mice exhibited osteoporotic features, thus suggesting that the receptor was involved in the signaling pathway of bone formation or bone resorption [21,22]. In contrast, the use of the GPR40 receptor agonist GW9508 reversed estrogen withdrawal osteoporosis induced by ovariectomy in mice [21,23]. Interestingly, GW9508 appears to have a dual effect of inhibiting osteoclastogenesis while promoting osteogenesis. At an appropriate concentration, GW9508 promoted apoptosis of osteoclast precursors [24] while it enhanced the proliferation of osteoblast precursors [16]. Mechanistically, GW9508 inhibited osteoclast differentiation by suppressing the NF-kB signaling pathway and the expression of the NFATc1 transcription factor [22,25], and it may have promoted osteogenic differentiation through the Wnt/β-catenin signaling pathway [23] and Ras-ERK1/2 cascade [26].

In summary, GW9508 has been demonstrated to have a systemic bone-inducing ability in a series of in vitro and in vivo studies; however, it remains unclear whether incorporating GW9508 into bone graft material can enhance local bone regeneration. Therefore, in this study, GW9508 was loaded into biomimetic extracellular matrix hydrogels and combined with a β-TCP/CaSiO_3_ scaffold to generate a novel bone graft material, and the biological effects of this osteogenic complex were investigated using in vitro studies and a rat cranial critical bone defect model. After the bone regeneration functions of the obtained osteogenic complex were confirmed, metabolomics analyses were conducted to explore the preliminary mechanism. The process of this research is illustrated in the Schematic Figure (Figure 1).

## 2. Materials and Methods

### 2.1. Materials

GW9508 was purchased from MedChemexpress (NJ, USA). Sprague-Dawley (SD) rats were purchased from Vital River (Beijing, China). SD rat mesenchymal stem cells (BMSCs) were purchased from Cyagen (Guangdong, China). α-Minimum essential medium Eagle (α-MEM), fetal bovine serum, and phosphate buffer solution (PBS) were purchased from Thermo Fisher Scientific (MA, USA). Xylenol orange, calcein, and dimethyl sulfoxide (DMSO) were purchased from Sigma-Aldrich (USA). Ascorbate, β-glycerophosphate, and dexamethasone were obtained from Sigma Aldrich (MO, USA). A penicillin-streptomycin solution was acquired from Biological Industries (Kibbutz Beit Haemek, Israel). A Nunc™ Lab-Tek™ II Chamber Slide™ system was purchased from Thermo Fisher Scientific (USA). Alginic acid sodium and hyaluronic acid were obtained from Sigma-Aldrich (USA). Collagen type I (cat #50201) was supplied by Ibidi (Munich, Germany). A HiScript III First Strand cDNA synthesis kit (R312) was purchased from Vazyme (China). ChamQ SYBR qPCR master mix (Q311) was purchased from Vazyme (Jiangsu, China). Methylene blue-basic fuchsin and safranin O-fast green stain kits were acquired from Servicebio (Hubei, China). ALP antibody (DF6225), RUNX2 antibody (AF5186), BMP-2 antibody (AF5163), and OPN antibody (AF0227) were purchased from Affinity Biosciences (OH, USA). A LIVE/DEAD™ cell imaging kit (R37601), anti-GPR40 (PA5-75351), anti-GPR120 (PA5-50973), Alexa Fluor™ 488 donkey anti-rabbit IgG (H+L) highly cross-adsorbed secondary antibody (A21206), and ProLong™ Gold antifade mountant with DAPI were supplied by Thermo Fisher Scientific (USA). A poly-HRP secondary antibody (PR30009) was purchased from Proteintech (IL, USA). DAB (ZLI-9017) was purchased from ZSGB-BIO (Beijing, China). A TRAP staining kit (294-67001) was purchased from Wako Co., Ltd. (Tokyo, Japan). A BCIP/NBT alkaline phosphatase color development kit was purchased from Beyotime (Beijing, China).

### 2.2. Fabrication of the Scaffolds and Measurement

The fabrication of the 3D-printed bio-ceramic scaffolds was based on previous studies [11,12]. Briefly, β-TCP and CaSiO_3_ particles were synthesized using a chemical precipitation method and then mixed with a dispersant and viscosifier to obtain a 3D-printing slurry. An extrusion 3D Bioplotter™ system (Regenovo, Hangzhou, China) was utilized to fabricate the scaffolds, and the 3D-printed scaffolds were calcined at 1100 °C. Surface phase characterization was performed using X-ray diffraction (XRD) from 5 to 70°, with a scanning step length and rate of 0.02° and 1°/min, respectively. The morphology of the scaffolds was observed by scanning electron microscopy (SEM, MERLIN, ZEISS) after a platinum sputtering treatment on the surface of the scaffolds.

### 2.3. Hydrogel Preparation, Fabrication of the GW-Osteogenic Complex, and High-Performance Liquid Chromatography (HPLC)

According to published methods of preparing hybrid hydrogels [14], the collagen alginate hydrogel was prepared by divalent calcium binding (CaCl_2_), in which calcium crosslinked the alginate. The mechanical properties of the hybrid collagen alginate hydrogel were enhanced by altering the calcium concentration. The hybrid inorganic-organic implantation scaffolds were obtained after integrating the 3D-printed β–TCP/CaSiO_3_ scaffolds with the hydrogel.

We mixed collagen type I with an alginate stock solution (50 mg/mL, low viscosity), hyaluronic acid (HA, 10 mg/mL), 10× DMEM, and cell culture medium, and then added a GW9508 (1 mM dissolved in DMSO) stock solution into the pre-hybrid hydrogel system. Finally, a CaCl_2_ solution was added to achieve a final concentration of 3 mg/mL for collagen, 5 mg/mL for alginate, 50 μM for GW9508, and 3.75 mM for CaCl_2_. The hydrogel was vortexed gently and dropped into the β–TCP/CaSiO_3_ scaffolds and then incubated with 5% CO_2_ for 20 min at 37 °C to form a GW9508-containing osteogenesis complex (GW-OC) or OC (without GW9508).

The experimental samples were prepared by adding 64 μg of GW9508 to the GW-OC (64 μg/mL) and 2 mL of PBS as the culture medium. Then, 1.8 mL of PBS was drawn out on days 1, 3, 5, 7, 9, 11, and 14 and replenished to 2 mL by adding PBS to the GW-OC. The experiment was performed using three independent replicates (n = 3). The standard sample preparation was to dissolve GW9508 directly in PBS. We prepared 64 μg/mL, 12.8 μg/mL, 2.56 μg/mL, and 0.512 μg/mL as standard samples. A Prominence LC-20A HPLC system (Shimadzu, Japan) equipped with a degasser, a quaternary gradient low-pressure pump, a CTO-20A column oven, and an SPD-M20A photodiode array detector (PDA) was used for analysis. The quantification of the target analytes was performed on an RP-18 end-capped column (Agilent ZORBAX StableBond 300 C18, 4.6 mm × 250 mm; 5 μm) with an injection volume of 500 μL following an optimized protocol based on previous studies (REF). The mobile phase was a mixture of water (A) and acetonitrile (B) containing 0.1% TFA, and the gradient elution steps were set as follows: 0–5 min, 10% B; 5–35 min, 10–90% B; 35–35.5 min, 90–10% B; 35.5–50 min, 10% B. The flow rate of the mobile phase was maintained at 1 mL/min, the column temperature was controlled at 30 °C, and the UV detection wavelength was set at 245 nm. Both experimental samples and standard samples were analyzed using HPLC with the above conditions.

### 2.4. Cell Culture In Vitro and Osteogenic Differentiation Measurements of GW9508

Rat BMSCs were cultivated under standard conditions (37 °C, 95% air: 5% CO_2_ *v*/*v*) in α-MEM supplemented with 10% fetal bovine serum and 1% *w*/*v* of penicillin-streptomycin. For osteogenic differentiation, cells were incubated up to 14 days in an osteogenic induction medium (OM) [27], which was obtained by adding 50 mg/mL ascorbate-2-phosphate, 10 mM β–glycerol phosphate, and 100 nM dexamethasone to the α–MEM. The OM medium was replaced every 3 days during the osteogenic differentiation of BMSCs. To confirm the optimum concentration of GW9508 for BMSCs, we used five different concentrations of GW9508, according to a previous study [24]. Groups were treated with GW9508 at 0 μM, 25 μM, 50 μM, 75 μM, or 100 μM. The medium with different concentrations of GW9508 was changed every 3 days. All of the groups were cultivated under standard conditions for 14 days.

### 2.5. Immunocytochemistry and Quantitative Real-Time Polymerase Chain Reaction (qRT-PCR) Assays

BMSCs cells were seeded in six-well plates (1 × 10^5^ cells/well) and cultured for 14 days. The osteogenic-related genes, including alkaline phosphatase (*Alp*), runt-related transcription factor 2 (*Runx2*), osterix (*Sp7*), and osteopontin (*Spp1*), were analyzed by RT-PCR, which was performed as described previously [28]. The relative mRNA expression of the target genes was calculated using the 2^−ΔΔCt^ method, and *Gapdh* and *β-actin* were used as reference genes. BMSCs were seeded in the GW-OC and OC (5 × 10^5^ cells/mL), cultured for 14 days, and then analyzed. The BMSC osteogenic-related genes were analyzed by the same method as described above. The primer sequences are shown in Table 1.

BMSCs were seeded in a Nunc™ Lab-Tek™ II Chamber Slide™ system with 1 × 10^5^ cells/mL and then cultivated in different concentrations of GW9508 for 14 days. Afterward, the cells were washed with a PBS solution and fixed with 4% paraformaldehyde for 20 min. After incubation with 0.5% Triton X-100 in PBS for 20 min, the samples were blocked in a PBS solution containing 5% BSA for 1 h. Next, the cells were incubated with the primary antibodies anti-ALP (1:200) and anti-RUNX2 (1:200) overnight at 4 °C. After primary antibody incubation, cells were washed with PBS and incubated with the secondary antibodies, which were conjugated to FITC at room temperature for 1 h. Finally, the chamber was removed, and the nuclei were stained with DAPI. The images were observed and collected with a laser scanning confocal microscope (Olympus FV3000, Japan), using ImageJ software to measure the average gray value of the ALP and RUNX2 protein bands.

BMSCs cells were seeded on the GW-OC or OC (5 × 10^5^ cells/mL) and cultured for 3 or 7 days. The GW-OC and OC were fixed with 4% paraformaldehyde for 30 min at room temperature and then washed three times by adding PBS. Antigen blocking was performed by incubating in 3% BSA with TritonX-100 at a final concentration of 0.1% at 37 °C for 1 h. Next, the GW-OC and OC were incubated at 4 °C overnight with the primary antibodies rabbit anti-GPR40 (1:200) and rabbit anti-GPR120 (1:200), and then washed with 1× PBST three times. The GW-OC and OC were incubated with their respective secondary antibodies (1:500) for 2 h at room temperature in a low-lit room. The cellular localization of GPR40 and GPR120 was observed with the use of a fluorescence microscope (Olympus FV3000, Japan), using ImageJ software to measure the average values of GPR40 (red) and GPR120 (green) (Appendix A).

### 2.6. Live/Dead Cell Staining and ALP Staining

BMSCs cells were seeded in the GW-OC or OC (5 × 10^5^ cells/mL) and cultured for 3 or 7 days. Two identical groups of the OC and GW-OC were prepared for the subsequent experiments. Live cell dye (FITC or GFP; excitation/emission 488/515 nm) and dead cell dye (Texas Red: excitation/emission 570/602 nm) were mixed according to the manufacturer’s instructions to create a 2X working solution. Then, 500 μL of the live/dead dye working solution was added to each sample and incubated for 15 min at 25 °C and immediately observed under a fluorescence microscope (Olympus FV3000, Japan), in which green fluorescence indicated live cells and red fluorescence identified dead cells. ImageJ-win 64 software was used to measure the cell viability ratio, using eight fields of view per well randomly selected for quantitative analysis (n = 8).

Another group of the OC and GW-OC was analyzed using ALP staining. The OC and GW-OC were washed with PBS three times (5 min each time), and fixed in 4% paraformaldehyde for 30 min. The washing step was then repeated. The BCIP/NBT stain stock solution was prepared according to the instructions. The samples were adequately covered with the solution and incubated in the dark at room temperature for 1 h. Then, ddH_2_O was added to stop the staining reaction.

### 2.7. Rat Calvarial Defect Model, Osteogenic Complex Implantation, and GW9508 Treatment

The in vivo study was approved by the Institutional Review Board and Animal Care Committee of Guangzhou Huawei Testing Co., Ltd. (approval number: 202208006), and the samples were further analyzed at the First Affiliated Hospital of Sun Yat-sen University. Twenty eight-week-old male SD rats weighing 280 ± 10 g were housed, two per cage, in a temperature-controlled room (temperature, 22.0 ± 1.0 °C; humidity, 40–70%) with light from 8 AM to 8 PM. The rats were allowed free access to water and food. Twenty rats were randomly divided into four groups: (1) control group with four rats, (2) GW9508 gavage (GW) group with four rats, (3) osteogenic complex (OC) group, and (4) GW9508 osteogenic complex (GW-OC) group. The remaining 12 rats were shared between the last two groups.

On the day of surgery, all of the animals were anesthetized with isoflurane (induction 3–5%; maintenance 1.5% in air). After cleaning and sterilizing the skin, a longitudinal incision of approximately 2.5 cm in length was made in the middle of the skull to cut the skin and subcutaneous tissue in turn. Next, two symmetrical full-thickness circular defects of 5 mm in diameter were made by an electric trephine drill in the parietal area of the rat cranium. Then, among the remaining 12 rats, the right calvarial defect sites were filled with the GW-OC (circular defect, n = 8), and the OC (circular defect, n = 8) was implanted in the left calvarial defect site. The group without any implantation was the control group (circular defect, n = 8). The GW group (circular defect, n = 8) was gavaged with GW9508 (dissolved in corn oil) at a dosage of 8 mg/kg of body weight three times per week for 4 weeks.

The animals were injected with streptomycin (0.125 mg) and penicillin (0.1 mg) once a day for 3 days after the operation. Four weeks after the operation, all of the rats were euthanized. Paraformaldehyde was perfused into the entire body, and all of the rat skulls were dissected and stored in 4% paraformaldehyde at 4 °C and then subjected to microcomputed tomography (micro-CT) scanning and histological analysis.

### 2.8. Micro-CT Scanning and Analysis

Two weeks after the operation, all of the animals in this study were anesthetized with 1% pentobarbital (0.4 mL/100 g) via intraperitoneal injection. Then, a SkyScan1276 Micro-CT system (Bruker, MA, USA) at 85 kV, 200 μA, and 1 mm aluminum filtration was applied to scan the calvarial defect sites. Four weeks after the operation, the dissected samples were also scanned in the same manner. The results were reconstructed with NRecon software (Bruker, Belgium). Three-dimensional analyses were carried out using CTvox software, which displayed the top view and general view at the 2W/4W time points. Additionally, the bone volume:tissue volume ratio (BV:TV), bone mineral density (BMD), trabecular thickness (Tb. th), and trabecular number (Tb. n) were calculated by CTAn analysis software (Bruker, Belgium). The operators and researchers conducting the micro-CT analysis were both blinded to the treatments associated with the samples.

### 2.9. Histological Assessment

One week after the operation, the GW-OC group and OC group received an intraperitoneal injection with xylenol orange (90 mg/kg). One week before all of the rats were sacrificed, the GW-OC group and OC group received an intraperitoneal injection with calcein (15 mg/kg). After the rats were sacrificed and the samples were scanned using micro-CT, four pairs of GW-OC and OC group cranial bone samples were prepared for hard-tissue sectioning. Then, sections were stained with methylene blue-basic fuchsin to observe the microarchitecture, while blank sections were analyzed under LSCM using calcein-xylenol orange fluorescent double-labeling, which showed the deposition rate of calcium between the GW-OC and OC groups.

Except for hard-tissue sectioned samples, all of the other cranial bone samples were decalcified in ethylene diamine tetraacetic acid (EDTA) for 6 weeks. After dehydration by a graded alcohol series and embedded in paraffin, the samples were cut into 8-μm sections for staining. The sections were stained with safranin O-fast green staining according to the manufacturer’s protocol (Servicebio, Hubei, China). TRAP staining for sections of the calvarial defect of each group was performed using a standard protocol to identify activated osteoclasts (TRAP staining kit, Wako Co. Ltd., 294-67001). The images were observed and captured by an image-scanning microscope (Leica, Weztlar, Germany). The TRAP+ cells were obtained by counting the number of positive-stained cells in the bone defect region.

The osteogenic-related proteins, OPN and BMP-2, were analyzed by employing immunohistochemistry (IHC), which measured the local expression of OPN and BMP-2. The IHC method was performed as described previously [29]. A Kfbio slide scanner (Zhejiang, China) was used for imaging the tissue sections. The IHC toolbox plugin (NIH, MD, USA) of ImageJ was used to quantify the immunohistochemistry images.

### 2.10. Metabolomics and Analysis

For untargeted metabolomics of polar metabolites, extracts were analyzed using quadrupole time-of-flight mass spectrometry (Q-TOF-MS; Sciex TripleTOF 6600) coupled with hydrophilic interaction chromatography via electrospray ionization at Shanghai Applied Protein Technology Co., Ltd. LC separation was performed on an ACQUIY UPLC BEH amide column (2.1 mm × 100 mm, 1.7 µm particle size; Waters, Wexford, Ireland) using a gradient of solvent A (25 mM ammonium acetate and 25 mM ammonium hydroxide in water) and solvent B (acetonitrile). The mass spectrometer was operated in both negative and positive ionization mode. In the MS acquisition, the instrument was set to acquire over an *m*/*z* range of 60–1000 Da, and the accumulation time for the TOF MS scan was 0.20 s/spectra. The product ion scan was acquired using information-dependent acquisition (IDA) with the high-sensitivity mode selected. Metabolic identification information was obtained by searching the laboratory’s self-built database and integrating a public database (Human Metabolome Database, HMDB) based on the exact masses of the molecular ions. Through variable importance in the projection (VIP) filtering, combined with the secondary spectrum score and difference multiples screening, differential metabolites were screened. Compound identification of metabolites was performed using the MS/MS spectra with an in-house database established with available authentic standards. After normalizing to the total peak intensity (all of the detected metabolites were normalized), the processed data were imported into SIMCA-P and subjected to multivariate data analysis, including Pareto scaled principal component analysis (PCA) and orthogonal partial least squares discriminant analysis (OPLS-DA). PCA was used for observing the overall distribution of each group, and OPLS-DA was applied for selecting differential metabolites between groups. Variables with a VIP > 1 were considered to be differential components. Non-parametric tests (*p* < 0.05) were utilized for selecting differential components, and VIP > 1 and *p* < 0.05 combined were used as the standard of differential metabolites.

### 2.11. Statistic

All of the data analyses were performed using Prism software (Version 9.1.0). Data are presented as means ± standard deviations (SD). Statistical analysis was performed using one-way ANOVA. For all of the experiments, *p* < 0.05 was considered to be significant: * *p* < 0.05; ** *p* < 0.01; *** *p* < 0.001; **** *p* < 0.0001. The experiments were performed randomly, and the investigators were blinded during experiments and outcome assessment.

## 3. Results

### 3.1. Physicochemical Characterization of the 3D-Printed β-TCP/CaSiO_3_ Scaffold and the Drug Release Properties of the Col/Alg/HA Hydrogel

The SEM images of the 3D-printed bio-ceramic scaffold are shown in Figure 1A,B. The architecture of the scaffold was inter-connected and porous, with pores around 250–300 μm. Micropores were formed on the surface of the scaffolds after thermal treatment, and the 3D-printed bio-ceramic scaffolds with macro-micro hierarchical pores were fabricated. The phases of the 3D-printed bio-ceramic scaffolds were confirmed to be β-TCP (JCPDS 00-006-0426) and CaSiO_3_ (JCPDS 00-003-0626) using XRD (Figure 1C).

According to the specification of GW9508, the peak of GW9508 was detected by HPLC when the UV detection wavelength was 245 nm. Based on the experimental conditions in the method section, at 29 min of HPLC processing, a single peak appeared in all of the groups, indicating that the detected peak area was GW9508 (Figure 2A). The standard curve and its regression equation Y = 9263*X + 586 (Figure 2B) was obtained by the relationship between the standard sample (*X*-axis) and the area under the peak (*Y*-axis) with known concentrations. The peak area of the experimental sample was detected by HPLC, and the GW9508 concentration of the experimental sample at different time points was obtained through a regression equation calculation (Figure 2C). The results showed that the GW9508 concentration of the experimental samples gradually decreased as time increased, confirming that the mass of the released GW9508 gradually decreased. By calculating the mass of GW9508 released on days 1, 3, 5, 7, 9, 11, and 14, the accumulated drug release curve of GW9508 was found (Figure 2D). The results showed that the accumulated release ratio of the GW-OC was about 60% at 14 days. In vitro results confirmed that the GW-OC exhibited a sustained release of GW9508, providing a basis to conduct in vivo GW-OC experiments.

### 3.2. GW9508 Promotes Osteogenic Differentiation of Rat BMSCs In Vitro

Previous studies have shown that GPR40 and GPR120 are expressed in different cell types, including osteoblasts and osteoclasts, and we have also confirmed their protein expression on rat BMSCs by immunofluorescence (Appendix A). This is the rationale for the use of GPR40 and GPR120 as activators of GW9508 to influence the differentiation of rat BMSCs. To verify the in vitro osteogenic effect of GW9508 and to determine the optimal drug concentration, BMSCs were incubated with different concentrations of GW9508 (0 μM, 25 μM, 50 μM, 75 μM, or 100 μM) for 14 days, and OM was used as a positive control. Immunofluorescence observed under a confocal microscope demonstrated that the 50 μL-treated group had the highest fluorescence expression intensity of osteogenesis-related proteins (ALP and RUNX2) (Figure 3A). When analyzed by ImageJ, the 50 μM group had significantly higher protein bands of ALP and RUNX2 among the five groups (Figure 3B,C). Moreover, qRT-PCR (Figure 3D–G) revealed that the 50 μM treatment significantly increased the expression of osteogenic genes, including *Alp*, *Runx2*, *Sp7*, and *Spp1*. In addition, the expression of osteogenic genes in the GW9508 groups was lower than that of the positive control groups, which indicated that GW9508 induced a moderate osteogenic differentiation effect on BMSCs. A repeated qRT-PCR analysis using *β-actin* as the reference gene also showed the same experimental trends (Appendix A). These results confirmed that 50 μM of GW9508 was the optimized concentration to facilitate osteogenic differentiation, and this concentration was used in follow-up in vivo experiments.

### 3.3. The GW-OC Promotes Osteogenic Differentiation of BMSCs In Vitro

The above experiment showed that GW9508 promoted the osteogenic differentiation of BMSCs in vitro, but cell growth and differentiation of BMSCs in vivo in GW-OC and OC still needed to be confirmed. First, we used live/dead staining to verify the living cells in the OC and GW-OC (Figure 4A). From the quantitative results of day 3 and day 7 (Figure 4B), the live:dead cell ratio was close to 100%, and there was no significant difference between the OC and GW-OC groups. Using ALP staining, the ALP expression of BMSCs in the OC and GW-OC was detected (Figure 4C). The staining results showed that the staining depth of the GW-OC containing GW9508 was much deeper than that of the OC without GW9508, thus indicating that the GW-OC had a better ability to promote osteogenic differentiation of BMSCs. Finally, qRT-PCR was used to compare the expression of osteogenic-related genes in BMSCs using the OC and GW-OC. The results demonstrated that within 14 days, the expression levels of *Alp*, *Runx2*, *Sp7*, and *Spp1* in BMSCs in the GW-OC increased (Figure 4D), thus confirming that the GW-OC promoted osteogenic differentiation of BMSCs in vitro.

### 3.4. Systemic GW9508 Gavage and Local Delivery by the GW-OC Promotes Bone Repair in a Rat Critical-Size Bone Defect Model

We next examined whether local delivery or systemic treatment using GW9508 promoted bone regeneration in a rat calvarial bone defect model. The control group and GW group rats received 5 mm calvarial bone defects, while the OC and GW-OC groups received implants at the sites of the bone defects (Figure 5A). All of the animals underwent satisfactory anesthesia, proper surgical handling, and hematischesis during the operations to ensure postoperative recovery. No wound infections or deaths were observed throughout the in vivo experiment. There was no significant difference in the behavior and activities of the rats compared with before the operation.

At 2 and 4 weeks after the operation, bone regeneration was evaluated by micro-CT. The 3D reconstructed images showed the dynamic healing processes of the bone (Figure 5B). The control group, without GW9508 gavage or implantation, showed the minimum new bone formation among the four groups. In comparison, new bone formation was observed in the GW group as the healing time increased. For the implantation groups, the new bone growth of the GW-OC group was more significant than that of the OC group. Consistently, the micro-CT analysis demonstrated that healing proceeded much more quickly in the GW-OC group and GW group than in the other groups. The quantitative analysis of the microarchitecture parameters of the regenerated bone showed a similar trend in the BV:TV ratio (Figure 5C), BMD (Figure 5D), Tb. th (Figure 5E), and Tb. n (Figure 5F). The control group showed significantly less bone regeneration, while the GW-OC group presented the best radiological signs among the four groups. There was less new bone regeneration in the GW group than in the GW-OC group.

Consistent with the micro-CT results, the histomorphometric analysis confirmed increased mineralization at the calvarial bone site of the GW-OC group. The hard tissue sections were characterized for their morphology through methylene blue-basic fuchsin staining (Figure 6A). Moreover, calcein-xylenol orange fluorescent double-labeling further showed a higher bone formation rate induced by the GW-OC (Figure 6A). Calcium deposition between the red line (xylenol orange label) and the green line (calcein label) shows the new bone formed within 2 weeks. Additionally, measurements of the fluorescent double-labeling distance showed significant differences between the GW-OC group and OC group (Figure 6A).

To analyze the subtle changes in bone microstructure at the histological level, we performed safranin O-fast green staining in each group. Mature bone was dyed blue, and the newly formed tissue in the calvarial bone defect was dyed red with safranin O-fast green staining (Figure 6B). The images showed that the trabecular bone mass in the GW-OC group and GW group were evidently higher than that of the OC group and control group, which was consistent with the results of micro-CT morphometry. Furthermore, the quantitative measurements of the newly formed tissue area showed significant differences between the GW-OC, GW, OC, and control groups (Figure 6B). The results of the osteogenic efficiencies were further confirmed by immunohistochemistry.

In the process of local bone formation, the activity of osteoclasts is suppressed when the activity of osteoblasts is active. We sought to demonstrate whether GW9508 promoted local osteogenesis by inhibiting osteoclast activity. The cranial defect sections (Figure 6C) showed that osteoclast activity of bone resorption in the control group and OC group were significantly higher than that in the GW group and GW-OC group (Figure 6C), which was confirmed by TRAP+ cell counting. This result was consistent with safranin O-fast green staining and verified that the suppression of osteoclast activity may be promoting local osteogenesis.

It is known that osteopontin (OPN) and bone morphogenetic protein-2 (BMP-2) play an important role in the formation and development of bone; thus, we next explored the protein expression of OPN and BMP-2 in each group by immunohistochemical staining (Figure 7A,B). The results showed that OPN and BMP-2 were expressed to the highest extent in the GW-OC group and were slightly higher than the levels in the GW group. The control group showed the lowest expression. The quantitative analysis of the average optical density values further confirmed the observed results (Figure 7C,D).

Overall, the results above suggested that the GW-OC implantation and GW9508 treatment promoted bone regeneration in the defect areas. Notably, the local application of GW-OC was more effective than systemic GW9508 gavage treatment.

### 3.5. GW9508 Drives Osteogenesis by Regulating Endogenous Metabolites

To interpret how GW9508 promoted changes in endogenous metabolites in BMSCs, the obtained metabolites were analyzed by untargeted metabolomics liquid chromatography-mass spectrometry (LC-MS). The mass spectrometer was operated in both negative and positive ion modes. A schematic representation of all detected metabolite components is shown in Figure 8A. The top three metabolites were carboxylic acids and derivatives (26.947%), fatty acyls (8.402%), and glycerophospholipids (7.582%). PCA revealed an excellent separation between the control group and the GW group (Figure 8B). We applied supervised OPLS-DA to evaluate model stability. We obtained the positive ion mode (R2Y = 0.9556; Q2 = −0.2229) and negative ion mode (R2Y = 0.9679; Q2 = −0.043) results, which were reliable and without overfitting (Figure 8C,D). OPLS-DA was used to analyze the altered levels of cellular metabolites after GW9508 treatment. We selected metabolites with VIP ≥ 1 and *p* < 0.05 as significant differential metabolites. After GW9508 treatment, 94 metabolites were upregulated, and 147 metabolites were downregulated in the positive mode, 6 metabolites were upregulated, and 37 metabolites were downregulated in the negative mode (Figure 8E,F). Subsequently, a heatmap was used to classify and display all detected metabolites, which showed that GW9508 changed the distribution of intracellular metabolites in BMSCs (Figure 8G). The most significant differential metabolite results from the positive and negative ion modes are shown in Figure 8H. These metabolites mainly included amino acids and their metabolites, glyceryl phosphatide, fatty acyls, and organic acids and their derivatives. To clarify the effects of GW9508 on BMSC metabolic pathways, a differential enrichment analysis of metabolite pathways was performed. Further metabolite enrichment analyses showed that GW9508 had significant effects on glutathione metabolism, taurine and hypotaurine metabolism, purine and pyrimidine metabolism, and amino acid metabolism in BMSCs (Figure 8I). This suggests that GW9508 exerts its osteogenesis effect likely by altering the metabolism of these metabolites.

## 4. Discussion

Here, we describe an osteogenic complex composed of a β-TCP/CaSiO_3_ scaffold, Col/Alg/HA hydrogel, and GW9508, which significantly facilitated the repair of critical bone defects in a rat cranial defect model. Critical-size bone defects refer to bone defects that regenerate no more than 10%, and these are one of the major challenges clinically, suggesting the important value of this study. The margins of bone defects have the ability to regenerate to some extent; however, critical-size bone defects are too large for the disrupted ends to connect with each other. Therefore, the best therapeutic strategy is to use a material with superior osteoconductivity as a bridge for connection while osteoconducting stem cells and progenitor cells using the osteogenic factors encapsulated in the material. The inorganic scaffold and hydrogel used in this study have demonstrated good biocompatibility, osteoconductivity, and some degree of osteoinductivity in previous studies [12,14], and the addition of GW9508 further enhanced the osteoinductivity of the osteogenic complex. Thus, in a rat cranial bone round defect model, the new bone was interspersed within the porous structure inside the osteogenic complex, forming a bridge across the entire bone defect area, and demonstrated an excellent bone repair effect. Further in-depth research on this osteogenic complex will bring more benefits to patients with critical-size bone defects.

The inclusion of GW9508 is the main innovation of this study. Previous studies on GW9508 were mainly focused on the anti-osteoporosis field, which found that GW9508 regulated both osteoblast and osteoclast functions through GPR40 and GPR120 receptors. In fact, GW9508 is able to control the cell fate of skeletal progenitor cells toward bone or cartilage. A previous important study [30] found that lipid scarcity in the tissue microenvironment caused skeletal progenitor cells to activate the FOXO transcription factor, which bound to the promoter of *Sox9* and increased its expression, causing chondrogenesis instead of osteogenesis. In contrast, the use of GW9508, an agonist of lipid acid receptors, prevented the increase of SOX9 induced by lipid scarcity. Furthermore, a local injection of GW9508 during fracture repair reduced cartilage formation in the bone callus.

As compared with previous studies, the present study is the first to investigate the application of GW9508 in local bone defect repair, including a systemic application and loading in the osteogenic complex, both of which showed significant bone repair effects. In terms of mechanism, previous studies focused on the transduction of molecular signaling pathways, such as osteogenesis-related β-catenin signaling pathways and osteoclastogenesis-related NF-kB signaling pathways. In the present study, we performed an untargeted metabolomics assay and found that GW9508 regulated various intracellular metabolic pathways related to osteogenesis, including glutathione metabolism, taurine and hypotaurine metabolism, thiamine metabolism, and purine and pyrimidine metabolism, which explains the osteogenic mechanism from a different perspective. Specifically, glutathione metabolism is essential for bone mass accumulation in osteoprogenitors and is a key regulator of PTH-induced bone anabolic responses to maintain biosynthesis [31]. It has also been reported that taurine and hypotaurine metabolism promoted bone mineral accumulation, which is a potential biological biomarker for the BMSC treatment of osteoporosis [32]. However, there is insufficient evidence that thiamine metabolism or purine and pyrimidine metabolism is involved in the osteogenesis process, which may suggest new research directions.

Although important preliminary discoveries are revealed by this study, there are also some limitations and shortcomings, and we will further investigate the remaining questions of this study in the future. First, in animal experiments, the drug concentration of GW9508 was determined based on the optimal concentration in cellular experiments. The optimal in vivo concentration of GW9508 remains unclear due to the large differences in the number of cells and the amount of fluids in the tissue microenvironment between in vitro and in vivo models. Fortunately, GW9508 did exert efficacy in promoting bone repair under the concentration of this study. We propose to implement drug concentration gradients and validate them in larger mammals to determine the optimal drug concentration for bone regeneration in the future. Second, in the metabolomics study, we used an untargeted metabolomics approach, which has the advantage of identifying more than 2000 differential metabolites, but the disadvantage of slightly less precise quantitative results. Third, the main purpose of this study was to confirm the effectiveness of the osteogenic complex loaded with GW9508 in repairing bone defects; however, several significantly affected metabolic signals suggested in metabolomics were not studied and validated in depth. Thus, in response to the clues found in untargeted metabolomics, we propose to investigate the mechanism of GW9508 through a more targeted metabolomic approach, which should be validated by more biochemical techniques in an effort to lay a solid theoretical foundation for the clinical translation of GW9508 and the osteogenic complex. Finally, due to the continued aging of the population, increasingly more patients with bone defects also suffer from osteoporosis. However, in this study, the bone defect animal model was in a normal bone mass state. In a future study, we will utilize an osteoporotic bone defect animal model and also improve the osteogenic complex for this specific pathological microenvironment.

## 5. Conclusions

In this study, the osteogenic complex was obtained by loading GW9508 on a hybrid inorganic-organic scaffold by integrating 3D-printing technology. In vitro and in vivo studies showed that the osteogenic complex enhanced the osteogenic differentiation of BMSCs, and facilitated bone regeneration in critical-size bone defects. These results may be related to several intracellular metabolism pathways, including purine and pyrimidine metabolism, amino acid metabolism, glutathione metabolism, and taurine and hypotaurine metabolism.

## Data Availability

The data presented in this study are available on request to the corresponding author.

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
