# Peer review of "3D-Printed Bioactive Scaffold Loaded with GW9508 Promotes Critical-Size Bone Defect Repair by Regulating Intracellular Metabolism"

_bioengineering, 2023, doi:10.3390/bioengineering10050535_

Round 1
Reviewer 1 Report
Authors in qPCR studies should perform tests for two housekeeping genes.
Author Response
Dear reviewer:
Thank you for your comments. We have studied the comments carefully and made revisions accordingly. The important changes are shown in red font in the main manuscript.
Thank you again for considering our revised manuscript. We are looking forward to a favorable decision.
Best wishes
Chun Liu
The First affiliated Hospital of Sun Yat-sen University
Sun Yat-sen University
Point-to-point Responses to Reviewers’ comments:
Review 1
Authors in qPCR studies should perform tests for two housekeeping genes.
Answer:Thank you for the suggestion, we have performed a repeated qPCR test using another housekeeping gene “β-actin”. The results show similar trends with the previous qPCR test using “GAPDH” as housekeeping gene, and we put the results in line 405-407 of the revised manuscript and the new supplementary figure 2.

Reviewer 2 Report
The manuscript entitled “3D printed bioactive scaffold loaded with GW9508 promotes critical-size bone defect repair by regulating intracellular metabolism” is well-designed and well-written. However, the current manuscript needs additional supporting experiments to meet the quality of Bioengineering. The reviewer recommends minor revision.
1. The GW9508 is encapsulated in the hydrogel, which may rapidly release from scaffolds. The authors should check the release of GW9508 in GW-OC scaffolds at different time intervals.
2. Did the author check or compare the osteogenic effect of different concentrations of GW9508 on mesenchymal stem cells with osteogenic differentiation media concentrations? Does GW9508 show a synergetic effect under osteogenic media conditions?
3. Authors should add mesenchymal stem cells' viability and osteogenic differentiation on OC and GW-OC scaffolds.
4. As GW9508 is a potent agonist of GPR40 and GPR120. The author should check the expression of both receptors in stem cells seeded on scaffolds.
Author Response
Dear reviewer:
Thank you for your comments. We have studied the comments carefully and made revisions accordingly. The important changes are shown in red font in the main manuscript.
Thank you again for considering our revised manuscript. We are looking forward to a favorable decision.
Best wishes
Chun Liu
The First affiliated Hospital of Sun Yat-sen University
Sun Yat-sen University
Review
The manuscript entitled “3D printed bioactive scaffold loaded with GW9508 promotes critical-size bone defect repair by regulating intracellular metabolism” is well-designed and well-written. However, the current manuscript needs additional supporting experiments to meet the quality of Bioengineering. The reviewer recommends minor revision.
- The GW9508 is encapsulated in the hydrogel, which may rapidly release from scaffolds. The authors should check the release of GW9508 in GW-OC scaffolds at different time intervals.
Answer:Thank you for the suggestion, we have performed a HPLC experiment test to the drug release property of the hydrogel, and the results are presented in line 373-387 of the revised manuscript and the new figure 2. In short, the GW9508 in the hydrogel were released gradually in 14 days and the accumulated drug release ratio at day 14 was about 60%.
- Did the author check or compare the osteogenic effect of different concentrations of GW9508 on mesenchymal stem cells with osteogenic differentiation media concentrations? Does GW9508 show a synergetic effect under osteogenic media conditions?
Answer:Thank you for the comment. We did not compare the osteogenic effect of GW9508 with different concentrations on mesenchymal stem cells under osteogenic media condition. Because previous study (DOI 10.1007/s11010-015-2626-5) had confirmed that GW9508 at 50 μM showed a synergetic effect under osteogenic media condition to promote osteogenic differentiation. Hence in this study, we focused on investgating whether GW9508 alone can promote the osteogenic differentiation of BMSCs and how strong the effect could be, with osteogenic medium as positive control.
- Authors should add mesenchymal stem cells' viability and osteogenic differentiation on OC and GW-OC scaffolds.
Answer:Thank you for the suggestion. We have added a live/dead cell staining experiment to confirm the cell viability on OC and GW-OC scaffolds, and we have also performed ALP staining and qPCR tests to investigate the osteogenic effects of OC and GW-OC scaffolds. All the results are presented in line 426-435 of the revised manuscript and in the new figure 4.
- As GW9508 is a potent agonist of GPR40 and GPR120. The author should check the expression of both receptors in stem cells seeded on scaffolds.
Answer:Thank you for the suggestion. We have added an immunofluorescence experiment to confirm the protein expression of GPR40 and GPR120 in the BMSCs seeded on scaffolds, and the results are presented in line 390-392of the revised manuscript and the new supplementary figure 1.

Reviewer 3 Report
3D printed scaffolds loaded with GW9508 have been tested as bone regenerating platforms using proper in vitro and in vivo models. Although this work has interest and it's well designed, an effort regarding to format, style and clarity should be done.
1. As 3D printing is the main technique to obtain these scaffolds, Introduction section needs to cover this topic, providing recent examples and describing correctly the state-of-art.
2. Drug relase dynamics from loaded scaffolds should be estimated through a quantitative assay, specially considering early and mid term conditions.
3.Although osteoblastic activity and matrix synthesis is well described, manuscript will improve its quality further describing osteoclastic behavior in critical size defects regeneration in presence of this drug, maybe through a specific immunostaining or another quantitative technique.
4. There are ortographical mistakes and other typos, such as capital letters (first sentence of the introduction section or in Schematic 1), confussions about GW9508 dossage (line 299, page 7) and/or other syntatic mistakes (i.e. IamgeJ). Please check carefully.
Author Response
Dear reviewer:
Thank you for your comments. We have studied the comments carefully and made revisions accordingly. The important changes are shown in red font in the main manuscript.
Thank you again for considering our revised manuscript. We are looking forward to a favorable decision.
Best wishes
Chun Liu
The First affiliated Hospital of Sun Yat-sen University
Sun Yat-sen University
Point-to-point Responses to Reviewers’ comments:
Review
3D printed scaffolds loaded with GW9508 have been tested as bone regenerating platforms using proper in vitro and in vivo models. Although this work has interest and it's well designed, an effort regarding to format, style and clarity should be done.
- As 3D printing is the main technique to obtain these scaffolds, Introduction section needs to cover this topic, providing recent examples and describing correctly the state-of-art.
Answer:Thank you for the suggestion. We had added a section in the introduction part about 3D printing in line 70-75 of the revised manuscript. “For rigid materials, 3D-printing techniques, which allow high precision, can be used to build a porous architecture of ceramic scaffolds[10], and they also provide the advantage of fabricating implants for personalized bone defects[11]. Thus, 3D-printed bioactive ceramic β-TCP/CaSiO3 scaffolds were fabricated based on the technological process developed in our previous studies[12-13], with mineral elements (Ca, P, and Si) that favored bone regeneration[14].”
- Drug relase dynamics from loaded scaffolds should be estimated through a quantitative assay, specially considering early and mid term conditions.
Answer:Thank you for the suggestion. We have added a HPLC experiment to test the drug release dynamic from loaded scaffold, and the results are presented in line 373-387 of the revised manuscript and the new figure 2.
- Although osteoblastic activity and matrix synthesis is well described, manuscript will improve its quality further describing osteoclastic behavior in critical size defects regeneration in presence of this drug, maybe through a specific immunostaining or another quantitative technique.
Answer:Thank you for the suggestion. We have performed a Tartrate-resistant acid phosphatase (TRAP) staining to describe osteoclastic behavior in critical size defects regeneration. And the results are showed in line 498-505 of the revised manuscript and the new figure 6C.
- There are ortographical mistakes and other typos, such as capital letters (first sentence of the introduction section or in Schematic 1), confussions about GW9508 dossage (line 299, page 7) and/or other syntatic mistakes (i.e. IamgeJ). Please check carefully.
Answer:Thank you for the suggestion. We are sorry for these mistakes, and we have checked the whole manuscript very carefully and revised the orthographical mistakes and syntactic mistakes including the mistakes mentioned above.

Reviewer 4 Report
This paper suggests that the use antagonist of fatty acid can promote osteogenesis and inhibit the osteoclastogenesis, especially in osteoporotic bone. The obtained results in osteogenesis are not convincing and experiments were performed in a healthy rat and not osteoporotic animal. For better analysis, the qRT-PCR results should be expressed following de ddCT methodology. In this representation, it is difficult to conclude on the osteogenic capabilities of the used drug. Furthermore, the semi-quantitative data obtained by Image J are not reliable for the protein quantification. No osteoclastogenesis was performed despite the initial hypothesis.
The durg was loaded in alginate based hydrogel. The in vitro release of the drug (kinetic + concentration) should be described before the in vivo experiments.
The hydrogel was therefore combined to 3D printed ceramic scaffold. The micro-CT and the histological results are in total contradiction. Indeed, the authors state that the micro-CT reveals an increase in the bone volume in the presence of bioceramic loaded with the drug. I was wondering if the harvested signal did not correspond to the ceramic scaffold instead of the de novo bone ? In the figure 4, it seems that the scaffold was colonized by a fibrous tissue instead of bone. The de novo done was limited to the few nodules formed in the dura side. Moreover, the number of rats is too small to support the statistical analysis.
Several typos should be corrected in the manuscript
Author Response
Dear reviewer:
Thank you for your comments. We have studied the comments carefully and made revisions accordingly. The important changes are shown in red font in the main manuscript.
Thank you again for considering our revised manuscript. We are looking forward to a favorable decision.
Best wishes
Chun Liu
The First affiliated Hospital of Sun Yat-sen University
Sun Yat-sen University
Point-to-point Responses to Reviewers’ comments:
Review
This paper suggests that the use antagonist of fatty acid can promote osteogenesis and inhibit the osteoclastogenesis, especially in osteoporotic bone. The obtained results in osteogenesis are not convincing and experiments were performed in a healthy rat and not osteoporotic animal. For better analysis, the qRT-PCR results should be expressed following de ddCT methodology. In this representation, it is difficult to conclude on the osteogenic capabilities of the used drug. Furthermore, the semi-quantitative data obtained by Image J are not reliable for the protein quantification. No osteoclastogenesis was performed despite the initial hypothesis.
The durg was loaded in alginate based hydrogel. The in vitro release of the drug (kinetic + concentration) should be described before the in vivo experiments.
The hydrogel was therefore combined to 3D printed ceramic scaffold. The micro-CT and the histological results are in total contradiction. Indeed, the authors state that the micro-CT reveals an increase in the bone volume in the presence of bioceramic loaded with the drug. I was wondering if the harvested signal did not correspond to the ceramic scaffold instead of the de novo bone ? In the figure 4, it seems that the scaffold was colonized by a fibrous tissue instead of bone. The de novo done was limited to the few nodules formed in the dura side. Moreover, the number of rats is too small to support the statistical analysis.
Several typos should be corrected in the manuscript
We thank the reviewer for the comments, below are our replies.
- Reply for “The obtained results in osteogenesis are not convincing and experiments were performed in a healthy rat and not osteoporotic animal.” :
Answer:We appreciate your comment. In the introduction part of previous version, we have emphasized too much on osteoporosis and the balance between osteoblasts and osteoclasts. In fact, we focused on the therapeutic effect of scaffolds on the repair of critical bone defects, and therefore healthy rats were used. We have made major revision of the introduction part.
- Reply for “better analysis, the qRT-PCR results should be expressed following de ddCT methodology.”:
Answer:Thank you for the comment. Actually, relative mRNA expression of target genes was calculated using the 2-ΔΔCt method, which had been mentioned in the Materials and Methods part (line 201-202 of the revised manuscript ).
- Reply for “No osteoclastogenesis was performed despite the initial hypothesis.”:
Answer:Thank you for the suggestion. We have added a Tartrate-resistant acid phosphatase (TRAP) staining to describe osteoclastic behavior in critical size defects regeneration. And the results are showed in line 498-505 of the revised manuscript and the new figure 6C.
- Reply for “The drug was loaded in alginate based hydrogel. The in vitro release of the drug (kinetic + concentration) should be described before the in vivo experiments.”:
Answer:Thank you for the advice. We have added a HPLC experiment to test the drug release dynamic from loaded hydrogel, and the results are presented in line 373-387 of the revised manuscript and the new figure 2.
- Reply for “The micro-CT and the histological results are in total contradiction. Indeed, the authors state that the micro-CT reveals an increase in the bone volume in the presence of bioceramic loaded with the drug. I was wondering if the harvested signal did not correspond to the ceramic scaffold instead of the de novo bone ?”:
Answer:Thanks for the comments. We adjusted the voltage and exposure time during CT scanning to obtain the best contrast, and used 360 degree scanning to reduce artifacts to obtain the best signal for de novo bone imaging. In addition, the threshold range selection for tomographic reconstruction and binary selection was based on the existing literature and our previous similar experimental protocols (https://doi.org/10.1016/j.jmst.2022.07.016), which have been repeated and verified by us.
- Reply for “In the figure 4, it seems that the scaffold was colonized by a fibrous tissue instead of bone. The de novo done was limited to the few nodules formed in the dura side.”:
Answer:Thanks for the comments. In the original figure 4, The selection of slices in the GW group was not typical enough (not cut to the middle of the circular defect), and we have made changes in the new figure 6B. Although the overall area of new bone was indeed limited, there were significant differences between the groups. Besides, because we sacrificed animals at 1 month post operation, some immature osteoid might be indistinguishable from fibrous tissue.
- Reply for “Moreover, the number of rats is too small to support the statistical analysis.”:
Answer:Thanks for the comments. We created two bone defects in each rat skull, thus ensuring eight samples per group in most experiments (except for the hard tissue section OC and GW-OC groups, which had only four samples). This design reduced the sacrifice of experimental animals.

Round 2
Reviewer 3 Report
Major concerns have now been solved, and new results improved manuscript quality and impact.